# From Multilingual Complexity to Emotional Clarity: Leveraging Commonsense to Unveil Emotions in Code-Mixed Dialogues

**Shivani Kumar[1], Ramaneswaran S[2], Md Shad Akhtar [1], Tanmoy Chakraborty[3],**
[1]IIIT Delhi, India, [2]NVIDIA, India, [3]IIT Delhi, India
shivaniku@iiitd.ac.in, s.ramaneswaran2000@gmail.com, shad.akhtar@iiitd.ac.in,
tanchak@iitd.ac.in

## Abstract

Understanding emotions during conversation is a fundamental aspect of human communication, driving NLP research for Emotion Recognition in Conversation (ERC). While considerable research has focused on discerning emotions of individual speakers in monolingual dialogues, understanding the emotional dynamics in code-mixed conversations has received relatively less attention. This motivates our undertaking of ERC for code-mixed conversations in this study. Recognizing that emotional intelligence encompasses a comprehension of worldly knowledge, we propose an innovative approach that integrates commonsense information with dialogue context to facilitate a deeper understanding of emotions. To achieve this, we devise an efficient pipeline that extracts relevant commonsense from existing knowledge graphs based on the code-mixed input. Subsequently, we develop an advanced fusion technique that seamlessly combines the acquired commonsense information with the dialogue representation obtained from a dedicated dialogue understanding module. Our comprehensive experimentation showcases the substantial performance improvement obtained through the systematic incorporation of commonsense in ERC. Both quantitative assessments and qualitative analyses further corroborate the validity of our hypothesis, reaffirming the pivotal role of commonsense integration in enhancing ERC.

## 1 Introduction

Dialogue serves as the predominant means of information exchange among individuals (Turnbull, 2003). Conversations, in their various forms such as text, audio, visual, or face-to-face interactions (Hakulinen, 2009; Caires and Vieira, 2010), can encompass a wide range of languages (Weigand, 2010; Kasper and Wagner, 2014). In reality, it is commonplace for individuals to engage in informal conversations with acquaintances that involve a mixture of languages (Tay, 1989; Tarihoran and

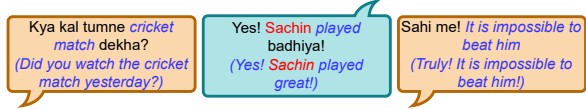

Figure 1: Example of a code-mixed dialogue between two speakers. Blue colour denote English words while red denotes proper noun.

Sumirat, 2022). For instance, two native Hindi speakers fluent in English may predominantly converse in Hindi while occasionally incorporating English words. Figure 1 illustrates an example of such a dialogue between two speakers in which each utterance incorporates both English and Hindi words with a proper noun. This linguistic phenomenon, characterized by the blending of multiple languages to convey a single nuanced expression, is commonly referred to as *code-mixing*.

While code-mixing indeed enhances our understanding of a statement (Kasper and Wagner, 2014), relying solely on the uttered words may not fully capture its true intent (Thara and Poornachandran, 2018). In order to facilitate better information assimilation, we often rely on various affective cues present in conversation, including emotions (Poria et al., 2019; Dynel, 2009; Joshi et al., 2017). Consequently, the task of Emotion Recognition in Conversation (ERC) (Hazarika et al., 2018b) has emerged and gained significant attention. ERC aims to establish a connection between individual utterances in a conversation and their corresponding emotions, encompassing a wide spectrum of possible emotional states. Despite the extensive exploration of ERC in numerous studies (Hazarika et al., 2018a; Zhong et al., 2019a; Ghosal et al., 2019; Jiao et al., 2019; Shen et al., 2020; Li et al., 2020; Jiao et al., 2020; Hazarika et al., 2021; Tu et al., 2022; Yang et al., 2022; Ma et al., 2022), the primary focus has been into monolingual dialogues, overlooking the prevalent practice of code-mixing. In this work, we aim to perform the task of ERC for code-mixed multi-party dialogues, thereby enabling the model-

ing of emotion analysis in real-world casual conversations. To the best of our knowledge, there is no previous work that deals with ERC for code-mixed conversations, leading to a scarcity of available resources in this domain. As a result, we curate a comprehensive dataset comprising code-mixed conversations, where each utterance is meticulously annotated with its corresponding emotion label.

The elicited emotion in a conversation can be influenced by numerous commonly understood factors that may not be explicitly expressed within the dialogue itself (Ghosal et al., 2020a). Consider an example in which the phrase "I walked for 20 kilometers" evokes the emotion of *pain*. This association stems from the commonsense understanding that walking such a considerable distance would likely result in fatigue, despite it not being explicitly mentioned. Consequently, capturing commonsense information alongside the dialogue context becomes paramount in order to accurately identify the elicited emotion. To address this, we propose incorporating commonsense for solving the task of ERC. However, the most popular commonsense graphs, such as ConceptNet (Speer et al., 2017) and COMET (Bosselut et al., 2019) are made for English, are known to work for the English language (Zhong et al., 2021; Ghosal et al., 2020b), and are not explored for code-mixed input. To overcome this challenge, we develop a pipeline to utilize existing English-based commonsense knowledge graphs to extract relevant knowledge for code-mixed inputs. Additionally, we introduce a clever fusion mechanism to combine the dialogue and commonsense features for solving the task at hand. In summary, our contributions are fourfold[1]:

1. We explore, for the first time, the task of **ERC for multi-party code-mixed conversations**.
2. We propose a **novel code-mixed multi-party conversation dataset**, E-MASAC, in which each discourse is annotated with emotions.
3. We develop `COFFEE`[2], **a method to extract commonsense knowledge** from English-based commonsense graphs given code-mixed input and fuse it with dialogue context efficiently.
4. We give a **detailed quantitative and qualitative analysis** of the results obtained and examine the performance of the popular large language models, including ChatGPT.

---

[1]The source code and dataset is present here: `https://github.com/LCS2-IIITD/EMNLP-COFFEE`.

[2]COmmonsense aware Fusion For Emotion rEcognition

## 2 Related Work

**Emotion recognition.** Earlier studies in emotion analysis (Ekman, 1992; Picard, 1997; Cowen and Keltner, 2017; Mencattini et al., 2014; Zhang et al., 2016; Cui et al., 2020) dealt with only standalone inputs, which lack any contextual information. To this end, the focus of emotion detection shifted to conversations, specifically ERC. While ERC was solved using heuristics and standard machine learning techniques initially (Fitrianie et al., 2003; Chuang and Wu, 2004; Li et al., 2007), the trend has recently shifted to employing a wide range of deep learning methods (Hazarika et al., 2018a; Zhong et al., 2019a; Li et al., 2020; Ghosal et al., 2019; Jiao et al., 2020; Hazarika et al., 2021; Shen et al., 2020; Poria et al., 2017b; Jiao et al., 2019; Tu et al., 2022; Yang et al., 2022; Ma et al., 2022).

**Emotion and commonsense.** Given the implicit significance of commonsense knowledge in the process of emotion identification, researchers have delved into the integration of commonsense for the purpose of emotion recognition. In scenarios involving standalone text, where the contextual information is relatively limited, studies propose the utilization of carefully curated latent commonsense concepts that can seamlessly blend with the text (Balahur et al., 2011; Zhong et al., 2021; Chen et al., 2022). However, in situations where the context of the text spans longer sequences, like dialogues, it becomes essential to capture it intelligently. Many studies explore the task of ERC with commonsense fusion using ConceptNet (Speer et al., 2017; Zhong et al., 2021), Atomic triplets (Sap et al., 2019; Nie et al., 2023), and the COMET graph (Bosselut et al., 2019; Ghosal et al., 2020b; Li et al., 2021).

**Emotion and code-mixing.** Existing research on emotion analysis for code-mixed language primarily focuses on standalone social media texts (Sasidhar et al., 2020; Ilyas et al., 2023; Wadhawan and Aggarwal, 2021) and reviews (Suciati and Budi, 2020; Zhu et al., 2022). While aspects such as sarcasm (Kumar et al., 2022a,b), humour (Bedi et al., 2023), and offense (Madhu et al., 2023) have been explored within *code-mixed conversations*, emotion analysis remains largely an uncharted territory with no relevant literature available, to the best of our knowledge. We aim to fill this gap by investigating the unexplored domain of ERC specifically for Hindi-English code-mixed conversations and introducing a novel dataset as well as a new model.

## 3 Problem Statement

Given, as input, the contextual utterances $(s_1, u_1), (s_2, u_2), \ldots, (s_m, u_{n-1})$ such that utterance $u_i$ is uttered by speaker $s_j$, the task of Emotion Recognition in Conversation aims to identify the emotion elicited in the target utterance $u_n$, uttered by the speaker $s_k$. All the discovered emotions come from a predefined set of emotion classes $E$. In this work, we consider the Eckman's emotion set which includes seven emotions – *anger*, *fear*, *disgust*, *sad*, *joy*, *surprise*, and *contempt*, along with a label for no emotion, i.e., *neutral*. Therefore, $e_i \in E$ where $E = \{anger, fear, disgust, sad, joy, surprise, contempt, neutral\}$.

## 4 The E-MASAC Dataset

A paucity of datasets exists for code-mixed conversations, making tasks on code-mixed dialogues scarce. Nevertheless, a recent dataset, MASAC (Bedi et al., 2023), compiled by extracting dialogues from an Indian TV series, contains sarcastic and humorous Hindi-English code-mixed multi-party instances. We extract dialogues from this dataset and perform annotations for the task of ERC to create E-MASAC. The resultant data contains a total of $8,607$ dialogues constituting of $11,440$ utterances. Data statistics are summarised in Table 1. Emotion distribution based on the annotations of the three sets[3] is illustrated in Figure 2.

**Emotion annotation.** Given, as input, a sequence of utterances forming a dialogue, $D = \{(s_1, u_1), (s_2, u_2), \cdots, (s_n, u_n)\}$, the aim here is to assign an appropriate emotion, $e_i$, for each utterance, $u_i$, uttered by speaker $s_j$. The emotion $e_i$ should come out of a set of possible emotions, $E$. Following the standard work in ERC for the English language, we use Eckman's emotions as our set of possible emotions as mentioned in Section 3, $E = \{anger, fear, disgust, sadness, joy, surprise, contempt, neutral\}$. Each emotion, along with its definition and example, is illustrated in Appendix A.1. We ask three annotators[4] $(a, b, c)$ to annotate each utterance, $u_i$, with the emotion they find most suitable for it, $e_i^a$ such that $e_i^a \in E$. A majority voting is done among the three annotations $(e_i^a, e_i^b, e_i^c)$ to select the final gold truth annotation,

| Set | #Dlgs | #Utts | Avg sp/dlg | Utt len | | Vocab len | |
|---|---|---|---|---|---|---|---|
| | | | | Avg | Max | English | Hindi |
| **Train** | 8506 | 8506 | 3.60 | 10.82 | 113 | | |
| **Val** | 45 | 1354 | 4.13 | 10.12 | 218 | 3157 | 14803 |
| **Test** | 56 | 1580 | 4.32 | 10.61 | 84 | | |
| **Total** | 8607 | 11440 | 12.05 | 31.55 | 415 | | |

Table 1: Data statistics for E-MASAC.

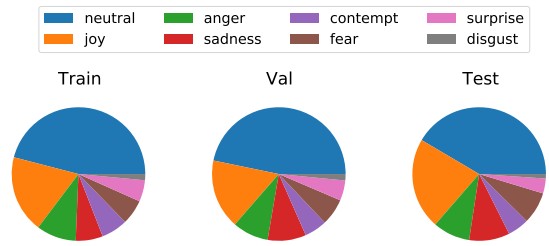

Figure 2: Emotion distribution for E-MASAC.

$e_i$. Any discrepancies are resolved by a discussion among the annotators; however, such discrepancies are rare. We calculate the inter-annotator agreement, using Kriprendorff's Alpha score (Krippendorff, 2011), between each pair of annotators, $\alpha_{ab} = 0.84$, $\alpha_{bc} = 0.85$, and $\alpha_{ac} = 0.85$. To find out the overall agreement score, we take the average score, $\alpha = 0.85$.

## 5 Proposed Methodology: COFFEE

As mentioned in Section 1, the manifestation of emotional concepts within an individual during a conversation is not solely influenced by the dialogue context, but also by the implicit knowledge accumulated through life experiences. This form of knowledge can be loosely referred to as commonsense. In light of this, we present an efficient yet straightforward methodology for extracting pertinent concepts from a given commonsense knowledge graph in the context of code-mixed inputs. Additionally, we introduce a clever strategy to seamlessly incorporate the commonsense features with the dialogue representation obtained from a backbone architecture dedicated to dialogue understanding. Figure 3 outlines our proposed approach, COFFEE while each of the intermediate modules is elucidated in detail below.

### 5.1 Dialogue Understanding Backbone (DUB)

For input containing long contextual history, such as a dialogue, it becomes crucial to capture and comprehend the entire progression leading up to the present statement. Consequently, an effective dialogue understanding architecture which gives us a concrete dialogue representation is required.

---

[3]We follow the original train-val-test split as is in MASAC

[4]The annotators are linguists fluent in English and Hindi with a good grasp of emotional knowledge. Their age lies between 25-30.

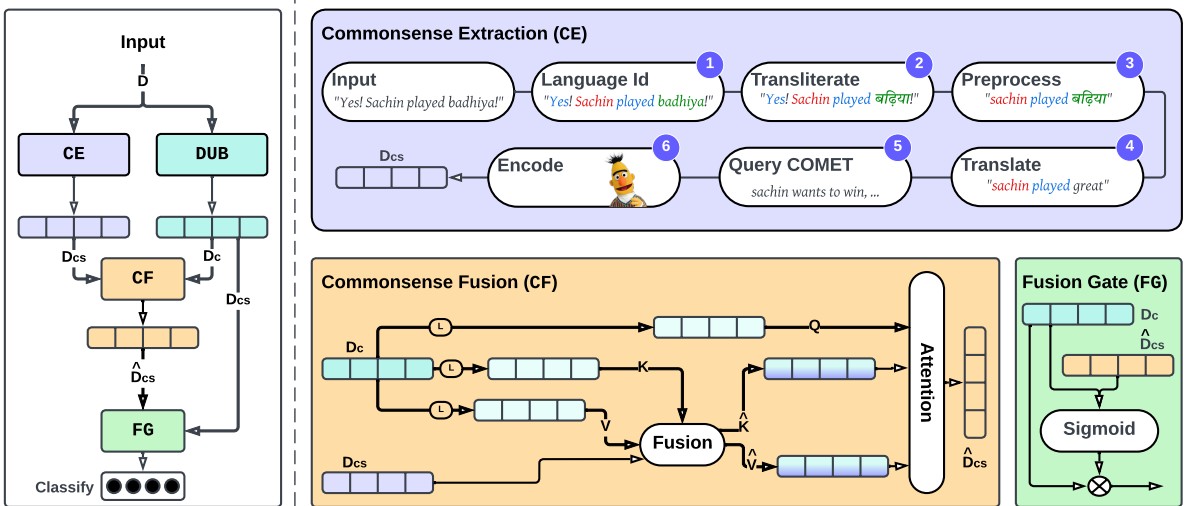

Figure 3: A schematic diagram of COFFEE. The Commonsense Extraction (CE) module takes a code-mixed input and provides a representation of the extracted commonsense information relevant to it. The commonsense information is fused with the representation obtained from a Dialogue Understanding Backbone (DUB) via the Commonsense Fusion (CF) and the Fusion Gate (FG) modules.

We use existing Transformer based architectures (c.f. Section 6.1) as our Dialogue Understanding Backbone, DUB. The given code-mixed dialogue $D$ goes through DUB to give us the contextual dialogue representation, $D_c$. Specifically, $D_c = \text{DUB}(D)$, such that $D_c \in \mathbb{R}^{n \times d}$ where $n$ is the maximum dialogue length, and $d$ is the dimensionality of the resulting vectors.

## 5.2 Commonsense Extraction (CE)

While the conversational context provides insights into the participants and the topic of the dialogue, the comprehension of implicit meanings within statements can be greatly facilitated by incorporating commonsense information. Therefore, in order to capture this valuable commonsense knowledge, we employ the COMET graph (Bosselut et al., 2019), which has been trained on ATOMIC triplets (Sap et al., 2019), to extract relevant commonsense information for each dialogue instance. However, it is worth noting that the COMET graph is pretrained using triplets in the English language, making it particularly effective for English inputs (Ghosal et al., 2020a). Given that our input consists of a mixture of English and Hindi, we have devised a specialized knowledge extraction pipeline to tackle this challenge. The entire process of obtaining commonsense knowledge for a given code-mixed textual input is shown in Figure 3 and is comprehensively explained below.

1. *Language Identification:* To handle the input dialogue $D$, which includes a mix of English and

| | |
|---|---|
| *oEffect* | The impact of input on the listeners. |
| *oReact* | The listeners' reaction to the input statement. |
| *oWant* | The listeners' desire after hearing the input. |
| *xAttr* | What the input reveals about the speaker. |
| *xEffect* | The speaker's desire after uttering the input. |
| *xIntent* | The speaker's objective in uttering the input. |
| *xNeed* | The speaker's needs according to the input. |
| *xReact* | The speaker's reaction based on the input. |
| *xWant* | The speaker's desire according to the input. |

Table 2: Commonsense effect-types returned by the COMET and their description.

Hindi words, the initial task is to determine the language of each word to appropriately handle different languages in the most suitable way.

2. *Transliteration:* The identified Hindi language words are transliterated to Devanagari script from roman script so that language-specific preprocessing can be applied to them.

3. *Text Processing:* The next step is to preprocess the text. This step involves converting text to lowercase and removal of non-ASCII characters and stopwords. The resultant text is considered important or *'topic specifying'* for the text.

4. *Translation:* Since COMET is trained for monolingual English, the query can only have English terms. Therefore, we translate the Devanagari Hindi *'topics'* back to romanised English.

5. *Querying COMET:* Finally, all the *'topics'* together are sent as a query to the COMET graph, and all possible relations are obtained.

COMET provides us with a vast array of effect-types corresponding to the input text. Specifically,

it provides us with information such as *oEffect*, *oReact*, *oWant*, *xAttr*, *xEffect*, *xIntent*, *xNeed*, *xReact*, *xWant*. Refer Table 2 for the description of each of these values. We carefully select the relevant attributes (c.f. Section 6.2) from the extracted pairs and encode them using the BERT model (Devlin et al., 2018). The representation obtained from BERT acts as our commonsense representation. Formally, $D_{cs} = \texttt{CE}(D)$, such that $D_{cs} \in \mathbb{R}^{m \times d}$ where $m$ is the length of the commonsense information, and $d$ is the vector dimension obtained from the BERT model. After we obtain the commonsense representation $D_{cs}$, we need to integrate it with the dialogue representation $D_c$. Consequently, we devise a sophisticated fusion mechanism as described in the following section.

### 5.3 Commonsense Fusion (CF)

Several studies discuss knowledge fusion, particularly in the context of multimodal fusion (Poria et al., 2017a), where the most successful approaches often employ traditional dot-product-based cross-modal attention (Bose et al., 2021; Ma and Ma, 2022). However, the traditional attention scheme results in the direct interaction of the fused information. As each fused information can be originated from a distinct embedding space, a direct fusion may be prone to noise and may not preserve maximum contextual information in the final representations. To address this, taking inspiration from Yang et al. (2019), we propose to fuse commonsense knowledge using a *context-aware attention mechanism*. Specifically, we first generate commonsense conditioned key and value vectors and then perform a scaled dot-product attention using them. We elaborate on the process below.

Given the dialogue representation $D_c$ obtained by a dialogue understanding backbone architecture, we calculate the query, key, and value vectors $Q$, $K$, and $V \in \mathbb{R}^{n \times d}$, respectively, as outlined in Equation 1 where $W_Q, W_K$, and $W_V \in \mathbb{R}^{d \times n}$ are learnable parameters, and $n$ and $d$ denote the maximum sequence length of the dialogue and dimensionality of the backbone architecture, respectively.

$$[QKV] = D_c [W_Q W_K W_V] \quad (1)$$

On the other hand, with the commonsense vector, $D_{cs}$, we generate commonsense infused key and value vectors $\hat{K}$ and $\hat{V}$, respectively as outlined in Equation 2, where $U_k$ and $U_v \in \mathbb{R}^{d \times d}$ are learnable matrices. A scalar $\lambda \in \mathbb{R}^{n \times 1}$ is employed

to regulate the extent of information to integrate from the commonsense knowledge and the amount of information to retain from the dialogue context. $\lambda$ is a learnable parameter learnt using Equation 3, where $W_{k_1}, W_{k_2}, W_{v_1}$, and $W_{v_2} \in \mathbb{R}^{d \times 1}$ are trained along with the model.

$$\begin{bmatrix} \hat{K} \\ \hat{V} \end{bmatrix} = (1 - \begin{bmatrix} \lambda_k \\ \lambda_v \end{bmatrix}) \begin{bmatrix} K \\ V \end{bmatrix} + \begin{bmatrix} \lambda_k \\ \lambda_v \end{bmatrix} (D_{cs} \begin{bmatrix} U_k \\ U_v \end{bmatrix}) \quad (2)$$

$$\begin{bmatrix} \lambda_k \\ \lambda_v \end{bmatrix} = \sigma(\begin{bmatrix} K \\ V \end{bmatrix} \begin{bmatrix} W_{k_1} \\ W_{v_1} \end{bmatrix} + D_{cs} \begin{bmatrix} U_k \\ U_v \end{bmatrix} \begin{bmatrix} W_{k_2} \\ W_{v_2} \end{bmatrix}) \quad (3)$$

Finally, the commonsense knowledge infused vectors $\hat{K}$ and $\hat{V}$ are used to compute the traditional scaled dot-product attention.

$$\hat{D_c} = Softmax(\frac{Q\hat{K}^T}{\sqrt{d_k}})\hat{V} \quad (4)$$

### 5.4 Fusion Gating (FG)

In order to control the extent of information transmitted from the commonsense knowledge and from the dialogue context, we use a Sigmoid gate. Specifically, $g = [D_c \oplus \hat{D_c}]W + b$. Here, $W \in \mathbb{R}^{2d \times d}$ and $b \in \mathbb{R}^{d \times 1}$ are trainable parameters, and $\oplus$ denotes concatenation. The final information fused representation $\hat{D_c}$ is given by $\hat{D_c} = D_c + g \odot \hat{D_c}$. $\hat{D_c}$ is used to identify the emotion class for the input dialogue.

## 6 Experiments and Results

### 6.1 Dialogue Understanding Backbone

As we mentiond earlier, existing approaches for ERC predominantly concentrate on the English language. Nonetheless, we incorporate two state-of-the-art techniques for ERC using English datasets and leverage four established Transformer-based methodologies as our foundation systems to address the ERC task.

**BERT** (Devlin et al., 2018) is a pre-trained language model that utilizes a Transformer architecture and bidirectional context to understand the meaning and relationships of words in a sentence. **RoBERTa** (Liu et al., 2019) is an extension of BERT that improves its performance utilizing additional training techniques such as dynamic masking, longer sequences, and more iterations. **mBERT** [5] (multilingual BERT) is a variant

---

[5]https://huggingface.co/M-CLIP/M-BERT-Base-ViT-B

| | Model | Anger | Contempt | Disgust | Fear | Joy | neutral | Sadness | Surprise | Weighted F1 |
|---|---|---|---|---|---|---|---|---|---|---|
| Standard | BERT | 0.23 | 0.18 | 0.11 | **0.20** | 0.45 | 0.54 | 0.16 | 0.32 | 0.40 |
| | RoBERTa | 0.26 | 0.21 | 0.16 | 0.06 | 0.47 | 0.57 | 0.12 | 0.34 | 0.41 |
| | mBERT | 0.10 | 0.11 | 0.00 | 0.11 | 0.23 | 0.50 | 0.13 | 0.08 | 0.30 |
| | MURIL | 0.24 | 0.22 | 0.07 | 0.00 | 0.42 | 0.51 | 0.06 | 0.23 | 0.35 |
| | CoMPM | 0.10 | 0.12 | 0.00 | 0.00 | 0.44 | 0.57 | 0.02 | 0.00 | 0.35 |
| | DialogXL | 0.25 | 0.09 | 0.07 | 0.17 | 0.43 | 0.59 | 0.17 | 0.28 | 0.41 |
| COFFEE | BERT | 0.24 (↑0.01) | 0.2 (↑0.02) | 0.12 (↑0.01) | 0.19 (↓0.01) | 0.46 (↑0.01) | 0.56 (↑0.02) | 0.18 (↑0.02) | **0.35** (↑0.03) | 0.41 (↑0.01) |
| | RoBERTa | **0.29** (↑0.03) | **0.24** (↑0.03) | **0.18** (↑0.02) | 0.10 (↑0.04) | **0.49** (↑0.02) | **0.61** (↑0.04) | **0.18** (↑0.06) | 0.34 (↑ 0.00) | **0.44** (↑0.03) |
| | mBERT | 0.11 (↑0.01) | 0.13 (↑0.02) | 0.04 (↑0.04) | 0.12 (↑0.01) | 0.24 (↑0.01) | 0.51 (↑0.01) | 0.12 (↓0.01) | 0.10 (↑0.02) | 0.31 (↑0.01) |
| | MURIL | 0.26 (↑0.02) | 0.21 (↓0.01) | 0.10 (↑0.03) | 0.01 (↑0.01) | 0.46 (↑0.04) | 0.52 (↑0.01) | 0.08 (↑0.02) | 0.22 (↓0.01) | 0.37 (↑0.02) |
| | CoMPM | 0.11 (↑0.01) | 0.14 (↑0.02) | 0.02 (↑0.02) | 0.02 (↑0.02) | 0.45 (↑0.01) | 0.56 (↓0.01) | 0.03 (↑0.01) | 0.10 (↑0.01) | 0.36 (↑0.01) |
| | DialogXL | 0.26 (↑0.01) | 0.11 (↑0.02) | 0.10 (↑0.03) | 0.19 (↑0.02) | 0.44 (↑0.01) | 0.59 (↑ 0.00) | 0.20 (↑0.03) | 0.31 (↑0.03) | 0.42 (↑0.01) |
| CS | KET | 0.14 (↓0.15) | 0.11 (↓0.13) | 0.09 (↓0.09) | 0 (↓0.10) | 0.34 (↓0.15) | 0.41 (↓0.20) | 0.08 (↓0.10) | 0.19 (↓0.15) | 0.28 (↓0.16) |
| | COSMIC | 0.21 (↓0.08) | 0.18 (↓0.06) | 0.15 (↓0.03) | 0.03 (↓0.07) | 0.39 (↓0.10) | 0.49 (↓0.12) | 0.13 (↓0.05) | 0.27 (↓0.07) | 0.34 (↓0.10) |

Table 3: Performance of comparative systems with and without incorporating commonsense via COFFEE. Numbers in parenthesis indicate the corresponding performance gain over the non-commonsense (standard) version. The last two rows compare the performance of the best performing COFFEE model (RoBERTa) with other commonsense (CS) based ERC methods.

of BERT that is trained on a multilingual corpus, enabling it to understand and process text in multiple languages. **MURIL** (Khanuja et al., 2021) (Multilingual Representations for Indian Languages) is a variant of BERT specifically designed to handle Indian languages. **CoMPM** (Lee and Lee, 2022) is a Transformer-based architecture especially curated for ERC. It extracts pre-trained memory as an extractor of external information from the pre-trained language model and combines it with the context model. **DialogXL** (Shen et al., 2020) addresses multi-party structures by utilizing increased memory to preserve longer historical context and dialog-aware self-attention. It alters XL-Net's recurrence method from segment to utterance level to better represent conversational data. **KET** (Zhong et al., 2019b) or the Knowledge-Enriched Transformer deciphers contextual statements by employing hierarchical self-attention, while simultaneously harnessing external common knowledge through an adaptable context-sensitive affective graph attention mechanism. **COSMIC** (Ghosal et al., 2020b) is a framework that integrates various aspects of common knowledge, including mental states, events, and causal connections, and uses them as a foundation for understanding how participants in a conversation interact with one another.

Although BERT and RoBERTa are trained using monolingual English corpus, we use them for romanised code-mixed input, anticipating that fine-tuning will help the models grasp Hindi-specific nuances (c.f. Appendix A.2). To ensure a fair comparison, we also include multilingual models such as mBERT and MURIL in our analysis. Additionally, since we are dealing with the task of ERC, we consider two state-of-the-art baseline architectures

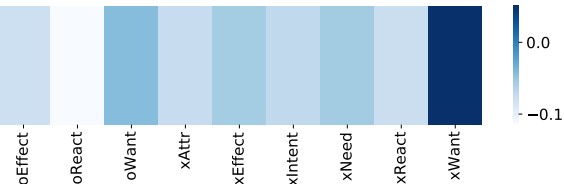

Figure 4: Correlation between different commonsense attributes with the emotion attribute.

in this domain for monolingual English dialogues, namely CoMPM and DialogXL and two state-of-the-art baseline that incorporates commonsense for ERC – KET, and COSMIC.

### 6.2 Experiment Setup and Evaluation Metric

The COMET graph gives us multiple attributes for one input text (c.f. Table 2). However, not all of them contributes towards the emotion elicited in the speaker. Consequently, we examine the correlation between the extracted commonsense attributes with emotion labels in our train instances. We use BERT to obtain representation for each commonsense attribute and find out their correlation with the emotion labels. We show this correlation in Figure 4. As can be seen, *'xWant'* is most positively correlated with the emotion labels, and *'oReact'* is most negatively correlated. Consequently, we select the attributes *'xWant'*, and *'oReact'* as commonsense. Further, for evaluating the performance, we select weighted F1 score as our metric of choice to handle the imbalanced class distribution of emotions present in our dataset (c.f. Figure 2).

### 6.3 Quantitative Analysis

Table 3 illustrates the results (F1-scores) we obtain for the task of ERC with and without using COFFEE

| | RoBERTa | Anger | Contempt | Disgust | Fear | Joy | neutral | Sadness | Surprise | Weighted F1 |
|---|---|---|---|---|---|---|---|---|---|---|
| | **Standard** | 0.26 | 0.21 | 0.16 | 0.06 | 0.47 | 0.57 | 0.12 | **0.34** | 0.41 |
| CS | **Concat** | 0.22 (↓0.04) | 0.19 (↓0.02) | 0.15 (↓0.01) | 0.04 (↓0.02) | 0.44 (↓0.03) | 0.52 (↓0.05) | 0.09 (↓0.03) | 0.31 (↓0.03) | 0.37 (↓0.04) |
| CS | **DPA** | 0.27 (↑0.01) | 0.21 (↑ 0.00) | 0.16 (↑ 0.00) | 0.08 (↑0.02) | 0.48 (↑0.01) | 0.59 (↑0.02) | 0.11 (↓0.01) | 0.33 (↓0.01) | 0.42 (↑0.01) |
| CS | **COFFEE$_{Eng}$** | 0.11 (↓0.15) | 0.09 (↓0.12) | 0.01 (↓0.15) | 0 (↓0.06) | 0.16 (↓0.31) | 0.24 (↓0.33) | 0.02 (↓0.10) | 0.11 (↓0.23) | 0.16 (↓0.25) |
| CS | **COFFEE$_{Hin}$** | 0.20 (↓0.06) | 0.15 (↓0.06) | 0.12 (↓0.04) | 0.02 (↓0.04) | 0.36 (↓0.11) | 0.53 (↓0.04) | 0.12 (↑0.00) | 0.29 (↓0.05) | 0.35 (↓0.06) |
| CS | **COFFEE$_{xW}$** | 0.26 (↑ 0.00) | 0.22 (↑0.01) | 0.15 (↓0.01) | 0.04 (↑0.02) | 0.47 (↑ 0.00) | 0.59 (↑ 0.02) | 0.16 (↑0.04) | 0.33 (↓0.01) | 0.42 (↑0.01) |
| CS | **COFFEE$_{oR}$** | 0.27 (↑0.01) | **0.24** (↑0.03) | 0.17 (↑0.01) | 0.07 (↑0.01) | 0.43 (↓0.04) | 0.59 (↑0.02) | **0.18** (↑0.06) | 0.33 (↓0.01) | 0.41 (↑ 0.00) |
| CS | **COFFEE** | **0.29** (↑0.03) | **0.24** (↑0.03) | **0.18** (↑0.02) | **0.10** (↑0.04) | **0.49** (↑0.02) | **0.61** (↑0.04) | **0.18** (↑0.06) | **0.34** (↑ 0.00) | **0.44** (↑0.03) |

Table 4: Ablation results comparing different fusion techniques for the best performing system (RoBERTa). Numbers in parenthesis indicate the performance gain over the non-commonsense (standard) version. Performance when only one of the matrix or embedding language is used for experimentation is also shown. (CS: Commonsense; DPA: Dot Product Attention; COFFEE$_{xW}$: COFFEE with only *xWant* attribute as commonsense knowledge; COFFEE$_{oR}$: COFFEE with only *oReact* attribute as commonsense knowledge).

| *oEffect* | *oReact* | *oWant* | *xAttr* | *xEffect* | *xIntent* | *xNeed* | *xReact* | *xWant* |
|---|---|---|---|---|---|---|---|---|
| 0.32 | 0.41 | 0.39 | 0.34 | 0.37 | 0.37 | 0.32 | 0.36 | **0.42** |

Table 5: Ablation results comparing different attibutes of commonsense when fused with RoBERTa using the COFFEE. The scores are weighted F1.

to incorporate commonsense knowledge. Notably, in the absence of commonsense, RoBERTa and DialogXL outperform the other systems. However, it is intriguing to observe that mBERT and MURIL, despite being trained on multilingual data, do not surpass the performance of BERT, RoBERTa, or DialogXL. We provide a detailed analysis regarding this in Appendix A.2. Further, when commonsense is included as part of the input using the COFFEE approach, all systems exhibit improved performance. The F1 scores corresponding to individual emotions show a proportional relationship with the quantity of data samples available for each specific emotion, as anticipated within a deep learning architecture. The *neutral* emotion achieves the highest performance, followed by *joy* and *surprise*, as these classes possess a greater number of data samples (see Table 2). Conversely, the minority classes such as *contempt* and *disgust* consistently obtain the lowest scores across almost all systems. Furthermore, we can observe from the table that the existing strategies of commonsense fusion perform poorly when compared with the COFFEE method. The loss in performance can be attributed to two aspects of the comparative system – KET uses NRC_VAD (Mohammad, 2018), which is an English-based lexicon containing VAD scores, i.e., valence, arousal, and dominance scores, to gather words for which knowledge is to be retrieved. Since our input is code-mixed with the matrix language as Hindi, using only the English terms makes the KET approach ineffective. In contrast, although COSMIC uses the COMET graph, it uses the raw representations obtained from the commonsense graph and concatenates them with the utterance representations obtained from the GRU architecture. Since we use the generated natural language commonsense with the smart fusion method, we hypothesize that our model is able to capture and utilize this knowledge effectively. Additionally, we performe a T-test on our results to check the statistical significance of our performance gain and obtained a p-value of $0.0321$ for our RoBERTa model which, being less than $0.05$, makes our results statistically significant.

### 6.4 Ablation Study

**Fusion methods** We investigate the effectiveness of COFFEE in capturing and incorporating commonsense information. To evaluate different mechanisms for integrating this knowledge into the dialogue context, we present the results in Table 4. Initially, we explore a straightforward method of concatenating the obtained commonsense knowledge with the dialogue context and passing it through the RoBERTa model. Interestingly, this simple concatenation leads to a decline in the performance of emotion recognition, suggesting that the introduced commonsense information may act as noise in certain cases. This outcome can be attributed to the inherent nature of some utterances, where external knowledge may not be necessary to accurately determine the expressed emotion. For instance, consider the sentence "Aaj me sad hun" (*"I am sad today"*), which can be comprehended without relying on commonsense information to identify the emotion as sadness. In such scenarios, enforcing additional information may disrupt the model's behavior, resulting in suboptimal performance. Conversely, by allowing the model the flexibility to decide when and to what extent to incorporate commonsense knowledge, as demon-

strated by the attention and COFFEE approaches, we observe an improvement in system performance, with COFFEE yielding the most favorable outcomes.

**Effect of language** In code-mixing, the input amalgamates two or more languages, often with one language being the dominant one, called the matrix language, while others act as embedding languages. The foundation of grammatical structure comes from the matrix language (Hindi in our case), and solely relying on the embedding language (English in our case) can lead to a decline in the model's performance. On the flip side, the embedding language plays a vital role in capturing accurate contextual details within the input. Therefore, confining ourselves to only the matrix language should also result in a drop in performance. To verify this hypothesis, the third and fourth row of Table 4 shows the performance of the COFFEE methodology, using the RoBERTa model, when we use only English (the embedding language) and only Hindi (the matrix language) in our input. The results reinforce our hypothesis, where the usage of only embedding language (English only) deteriorates the model performance extensively, while the sole use of matrix language (Hindi only) also hampers the performance when compared to the system that uses both the languages.

**COMET attributes** We explore the utilization of various COMET attributes as our commonsense information. The last three rows in Table 4 demonstrate the outcomes when we integrate the two most correlated attributes, *xWant* and *oReact* with the RoBERTa backbone model using COFFEE. It is evident that the individual consideration of these attributes does not significantly enhance the performance of ERC compared to when they are combined. Additionally, Table 5 presents the weighted F1 scores achieved by the RoBERTa model when each commonsense attribute is incorporated individually using COFFEE. These results align well with the observed correlation between the attributes and the corresponding emotion labels in Figure 4.

## 6.5 Qualitative Analysis

A thorough quantitative analysis, detailed in the previous section, revealed that the integration of commonsense knowledge enhances the performance of all systems under examination. However, to gain a deeper understanding of the underlying reasons for this improvement, we conduct a comprehensive qualitative analysis, comprising of

|  |  | **Predicted** | | | | | | | |
|  |  | An | Co | Di | Fe | Jo | Ne | Sa | Su |
|---|---|---|---|---|---|---|---|---|---|
| **Gold** | An | 26/33 | 5/5 | 8/4 | 2/9 | 13/20 | 86/62 | 1/4 | 1/5 |
|  | Co | 6/4 | 9/16 | 6/6 | 1/2 | 7/8 | 50/42 | 2/2 | 1/2 |
|  | Di | 2/6 | 1/3 | 5/2 | 0/0 | 1/1 | 7/4 | 0/0 | 1/1 |
|  | Fe | 9/13 | 2/5 | 3/4 | 1/3 | 17/19 | 76/55 | 12/17 | 2/6 |
|  | Jo | 3/8 | 3/4 | 2/2 | 1/9 | 159/171 | 162/139 | 15/8 | 4/8 |
|  | Ne | 21/32 | 14/18 | 2/7 | 2/9 | 76/81 | 496/450 | 21/27 | 24/32 |
|  | Sa | 5/11 | 7/6 | 5/3 | 0/1 | 19/26 | 90/71 | 28/35 | 1/2 |
|  | Su | 1/1 | 0/0 | 0/0 | 0/0 | 8/6 | 22/18 | 0/1 | 26/31 |

Table 6: Confusion matrices for ERC for the best performing RoBERTa model (without/with commonsense). (An: Anger; Co: Contempt; Di: Disgust; Fe: Fear; Jo: Joy; Ne: Neutral; Sa: Sadness; Su: Surprise).

confusion matrices and subjective evaluations.

### 6.5.1 Confusion Matrix

Given the superior performance of the RoBERTa model we conduct an examination of its confusion matrices with and without commonsense fusion, as shown in Table 6. We observe that the RoBERTa model with COFFEE integration achieves a higher number of true positives for most emotions. However, it also exhibits a relatively higher number of false negatives when compared with its standard variant, particularly for the *neutral* class. This observation suggests that the commonsense-infused model excels in recall but introduces some challenges in terms of precision, thereby presenting an intriguing avenue for future research. Additionally, we notice a heightened level of confusion between *neutral* and *joy* emotions, primarily due to their prevalence in the dataset. Both models, however, demonstrate the least confusion between the *disgust* and *surprise* emotions, indicating their distinguishable characteristics.

### 6.5.2 Subjective Evaluation

For the purpose of illustration, we select a single instance from the test set of E-MASAC and present, with it, the ground-truth and the predicted labels for the task of ERC for the best performing RoBERTa model with and without using the COFFEE approach in Table 7. It can be observed that the inclusion of commonsense knowledge in the model significantly reduces errors. Comparatively, the variant of RoBERTa that does not incorporate commonsense knowledge makes errors in 5 out of 9 instances, whereas the variant utilizing commonsense knowledge, using COFFEE, misclassifies only 3 utterances. Within the test set, numerous similar instances exist where the commonsense-infused variant outperforms its counterpart due to the implicit information embedded in the utterances.

| # | Speaker | Utterance | Emotion | | |
|---|---|---|---|---|---|
| | | | Gold | w/o CS | w CS |
| u1 | Maya | Khatam ho gaya Sahil it's over! *(It's over, Sahil!)* | sadness | sadness | sadness |
| u2 | Monisha | Mummyji, tissue paper ke aur 2 boxes hai, laati hun. *(Mummyji, I have two more boxes of tissue paper, I'll bring them.)* | neutral | joy | neutral |
| u3 | Sahil | Monisha, mom tissue paper ki baat nhi kar rhi hai... *(Monisha, mom is not talking about tissue paper...)* | neutral | sadness | neutral |
| u4 | Maya | My life! Meri zindagi! Khatam ho gayi hai. Can you imagine Sahil? Uss Rita se toh Monisha zyada achi hai. Can you imagine? *(My life! My life! It's over. Can you imagine, Sahil? Monisha is better than that Rita. Can you imagine?)* | sadness | sadness | sadness |
| u5 | Monisha | Mein kya itni buri hun mummy ji? *(Am I that bad, mummyji?)* | sadness | sadness | contempt |
| u6 | Maya | Haan beta. Lekin wo Rita! Oh my god! Saans leti hai toh bhi cheekh sunai deti hai. Jab logo ko pata chalega ke rosesh ne loudspeaker se shaadi ki hai?! *(Yes, dear. But that Rita! Oh my god! Even when she breathes, she makes a sound. When people find out that Rosesh got married through a loudspeaker?!)* | sadness | disgust | sadness |
| u7 | Monisha | Logo ko pata chal gaya mummyji... *( People found out, mummyji...)* | neutral | surprise | surprise |
| u8 | Maya | What do you mean?! *(What do you mean?!)* | fear | fear | contempt |
| u9 | Monisha | Wo sarita aunty ka phone aaya tha na... *(Sarita aunty called, right...)* | neutral | surprise | neutral |

Table 7: Actual and predicted emotions (using RoBERTa) for a dialogue having nine utterances from the test set of E-MASAC. Red-colored text represents misclassification.

## 6.6 ChatGPT and Code-mixing

Considering the emergence and popularity of Chat-GPT, it becomes imperative to conduct an analysis of it for the task of ERC in code-mixed dialogues. Although ChatGPT exhibits remarkable performance in a zero-shot setting across various tasks and scenarios, it is important to note its shortcomings, particularly when dealing with code-mixed input. To evaluate its performance, we extract instances from E-MASAC and engage ChatGPT in identifying the emotions evoked within the dialogues. To accomplish this, we construct a prompt that includes a potential set of emotions along with the code-mixed dialogue as input. Specifically, the prompt used is as follows:

*"Out of the following emotion set : {Anger, Contempt, Disgust, Fear, Joy, Neutral, Sadness, Surprise}, find out the emotion for the last utterance given the following conversation. <Conv>"*

While ChatGPT successfully discerned emotions in short and straightforward conversations, it faltered in identifying the appropriate emotion as the dialogue context expanded, sometimes even spanning more than three utterances. We present more details and example instances in Appendix A.3.

## 7 Conclusion

In this study, we investigated the task of Emotion Recognition in Conversation (ERC) for code-mixed dialogues, the first effort in its kind. We curate an emotion lad dialogue dataset for Hindi-English conversations and proposed a new methodology that leverages existing commonsense knowledge graphs to extract pertinent commonsense concepts for code-mixed inputs. The extracted common-sense is integrated into a backbone architecture through a novel fusion technique that uses context aware attention mechanism. Our findings indicated that the incorporation of commonsense features significantly enhances the performance of ERC, as evidenced both quantitatively and qualitatively.

## 8 Acknowledgements

The authors acknowledge the support of the ihub-Anubhuti-iiitd Foundation, set up under the NM-ICPS scheme of the DST. This work is also supported by Infosys Foundation.

## 9 Limitations

This work marks the inception of emotion identification in code-mixed conversations, opening up a plethora of research possibilities. One avenue for future studies is training the tokenizer specifically for code-mixed input, which can enhance the model's performance. Moreover, it is worth noting that our dataset comprises dialogues extracted from a situational comedy (sit-com), which may not encompass all real-world scenarios, potentially limiting the generalizability of our model to unseen situations. Consequently, future investigations can delve into diversifying the dataset to incorporate a broader range of contexts. Furthermore, conducting in-depth analysis to identify the optimal combination of commonsense attributes from COMET for ERC would shed further light on improving the system's performance. Although these avenues were beyond the scope of our current study, they present exciting prospects for future research.

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

## A  Appendix

### A.1  Emotion Labels

We create E-MASAC by utilising code-mixed dialogues from MASAC and asking annotators to annotate them with emotion labels. We consider the Eckman's emotions[6] as our choice of emotion labels due its prevalence in established English based ERC. This set contains seven emotions, namely *anger*, *fear*, *disgust*, *sadness*, *joy*, *surprise*, and *contempt*, along with a label for no emotion, i.e. *neutral*. Each emotion with their definition is illustrated in Table 8.

### A.2  Embedding Space for BERT

We employed various Transformer based methods as our dialogue understanding backbone to complete the task of ERC. Although RoBERTa is performing best for ERC, it is important to note that RoBERTa is pre-trained on English datasets (BookCorpus (Zhu et al., 2015) and English Wikipedia.). In order to explore how the representation learning is being transferred to a code-mixed setting, we analyse the embedding space learnt by the model before and after fine-tuning it for our task. We considered three random utterances from E-MASAC and created three copies of them- one in English, one in Hindi (romanised), and one without modification i.e. code-mixed. Figure 5 illustrates the PCA plot for the embeddings obtained for these nine utterance representations obtained by RoBERTa before and after fine-tuning on our task. It is interesting to note that even before any fine-tuning the Hindi, English, and code-mixed representations lie closer to each other and they shift further closer when we fine-tune our model. This phenomenon can be justified as out input is of romanised code-mixed format and thus we can assume that representations are already being captured by the pre-trained model. Fine-tuning helps us understand the Hindi part of the input.

### A.3  ERC for Code-mixed Dialogues Using ChatGPT

Given the rise and widespread adoption of expansive language models such as ChatGPT, it is crucial to undertake a comprehensive analysis of its capabilities in the context of ERC. While ChatGPT has demonstrated impressive performance in

---

[6]We consider its latest version picked from here: https://www.paulekman.com/universal-emotions/

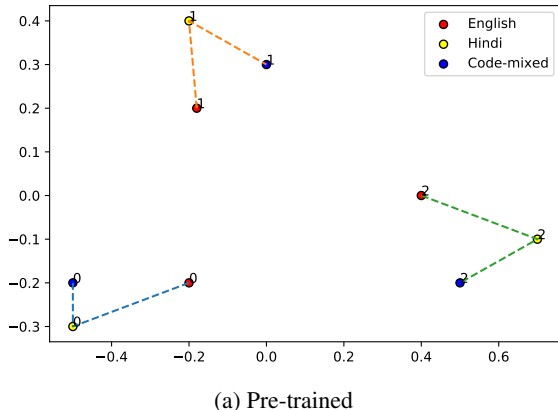

(a) Pre-trained

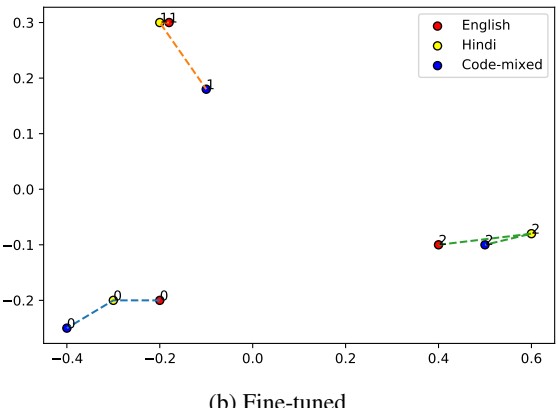

(b) Fine-tuned

Figure 5: Embedding space for RoBERTa before and after fine-tuning on emotion recognition in conversation.

zero-shot scenarios across diverse tasks, it is essential to acknowledge its limitations, especially when confronted with code-mixed input. To assess its efficacy, we select instances from the E-MASAC and task ChatGPT with the identification of emotions elicited within the dialogues. By subjecting ChatGPT to this evaluation, we gain insights into its effectiveness for the ERC task. To accomplish this, we construct a prompt that includes a potential set of emotions along with the code-mixed dialogue as input. Specifically, the prompt is as follows:

*"Out of the following emotion set : {Anger, Contempt, Disgust, Fear, Joy, Neutral, Sadness, Surprise}, find out the emotion for the last utterance given the following conversation. <Conv>"*
Although ChatGPT demonstrated proficiency in discerning emotions within concise and uncomplicated conversations, its performance waned when faced with the challenge of identifying the accurate emotion as the dialogue context extended, occasionally encompassing more than three utterances. Figure 6 shows four such instances and Table 9 shows their translation for easy understanding.

| Emotion | Description | Example |
|---|---|---|
| **Anger** | Arises when the target is blocked from pursuing a goal and/or treated unfairly | "Get out of my way!" |
| **Contempt** | Feeling of dislike for and superiority (usually morally) over another person, group of people, and/or their actions | "I'm better than you and you are lesser than me." |
| **Disgust** | Feeling of aversion towards something offensive | "Who put this dead mouse here?!" |
| **Joy** | Arising from connection or sensory pleasure | "That was so fun!" |
| **Fear** | Arises with the threat of harm, either physical, emotional, or psychological, real or imagined | "Is there no way out from here?" |
| **Sadness** | Resulting from the loss of someone or something important. What causes one's sadness varies greatly based on personal and cultural notions of loss | "My dog died two days ago" |
| **Surprise** | Arises when the target encounters sudden and unexpected sounds, movements, situations and actions | "I did not expected it to be this good!" |
| **Neutral** | When the target is experiencing none of the emotions | "Hey, what are you doing today?" |

Table 8: Eckman's emotions, their description, and examples.

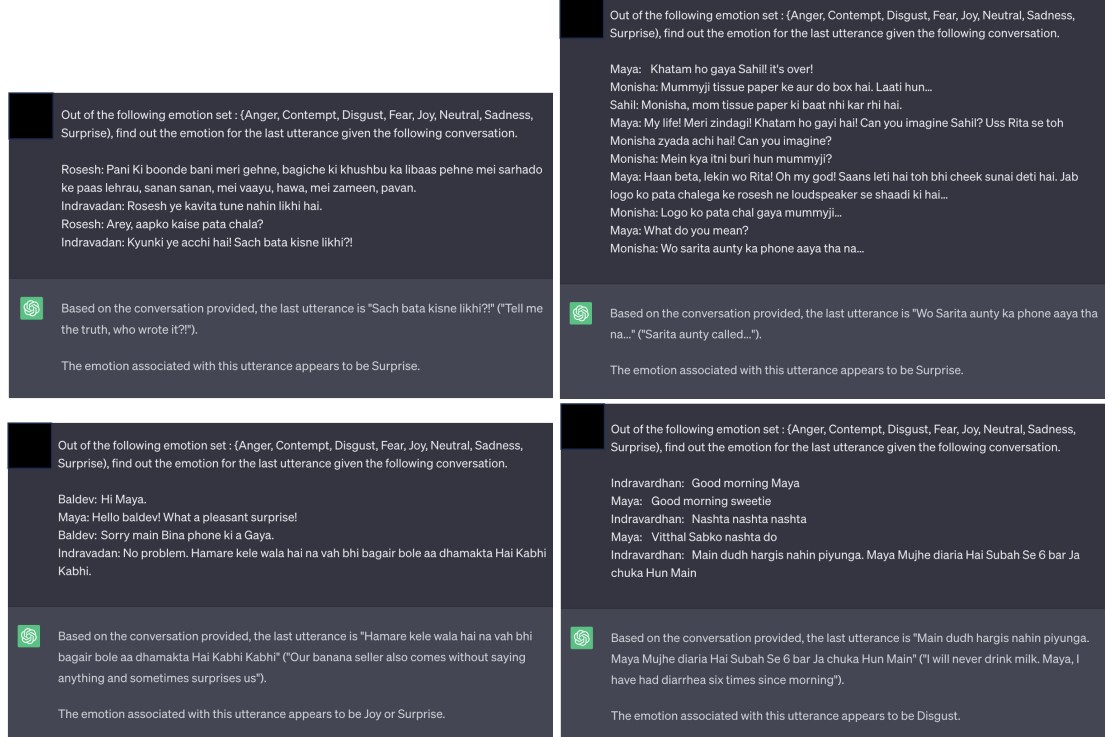

Figure 6: Screenshots of ChatGPT responses when prompted to provide emotions for the last utterance in the dialogue. The translations are given in Table 9.

| # | Code-mixed Dialogues | English Translation | ChatGPT Emotion |
|---|---|---|---|
| 1 | **Rosesh**: Pani Ki boonde bani meri gehne, bagiche ki khushbu ka libaas pehne mei sarhado ke paas lehrau, sanan sanan, mei vaayu, hawa, mei zameen, pavan. 
 **Indravardhan**: Rosesh ye kavita tune nahin likhi hai. 
 **Rosesh**: Arey, aapko kaise pata chala? 
 Indravardhan: Kyunki ye acchi hai! Sach bata kisne likhi?! | **Rosesh**: Water droplets have turned into my jewelry, I adorn the attire of the garden's fragrance, I sway near the borders, swaying gently. I am the air, the breeze, I am the earth, the wind. 
 **Indravardhan**: Rosesh, you didn't write this poem. 
 **Rosesh**: Oh, how did you know? 
 Indravardhan: Because it's good! Tell me the truth, who wrote it?! | Surprise |
| 2 | **Maya**: Khatam ho gaya Sahil! it's over! 
 **Monisha**: Mummyji tissue paper ke aur do box hai. Laati hun... 
 **Sahil**: Monisha, mom tissue paper ki baat nhi kar rhi hai. 
 **Maya**: My life! Meri zindagi! Khatam ho gayi hai! Can you imagine Sahil? Uss Rita se toh Monisha zyada achi hai! Can you imagine? 
 **Monisha**: Mein kya itni buri hun mummyji? 
 **Maya**: Haan beta, lekin wo Rita! Oh my god! Saans leti hai toh bhi cheek sunai deti hai. Jab logo ko pata chalega ke rosesh ne loudspeaker se shaadi ki hai... 
 **Monisha**: Logo ko pata chal gaya mummyji... 
 **Maya**: What do you mean? 
 Monisha: Wo sarita aunty ka phone aaya tha na... | **Maya**: It's over, Sahil! 
 **Monisha**: Mummyji, I have two more boxes of tissue paper. I'll bring them. 
 **Sahil**: Monisha, Mom is not talking about tissue paper... 
 **Maya**: My life! My life! It's over. Can you imagine, Sahil? Monisha is better than that Rita. Can you imagine? 
 **Monisha**: Am I that bad, mummyji? 
 **Maya**: Yes, dear. But that Rita! Oh my god! Even when she breathes, she makes a sound. When people find out that Rosesh got married through a loudspeaker?! 
 **Monisha**: People found out, mummyji... 
 **Maya**: What do you mean?! 
 Monisha: Sarita aunty called, right... | Surprise |
| 3 | **Baldev**: Hi Maya. 
 **Maya**: Hello baldev! What a pleasant surprise! 
 **Baldev**: Sorry main Bina phone ki a Gaya. 
 Indravardhan: No problem. Hamare kele wala hai na vah bhi bagair bole aa dhamakta Hai Kabhi Kabhi. | **Baldev**: Hi Maya. 
 **Maya**: Hello Baldev! What a pleasant surprise! 
 **Baldev**: Sorry, I came without calling. 
 Indravardhan: No problem. Our banana seller also comes without saying anything and sometimes surprises us. | Joy or Surprise |
| 4 | **Indravardhan**: Good morning Maya 
 **Maya**: Good morning sweetie 
 **Indravardhan**: Nashta nashta nashta 
 **Maya**: Vitthal Sabko nashta do 
 Indravardhan: Main dudh hargis nahin piyunga. Maya Mujhe diaria Hai Subah Se 6 bar Ja chuka Hun Main | **Indravardhan**: Good morning, Maya. 
 **Maya**: Good morning, sweetie. 
 **Indravardhan**: Breakfast, breakfast, breakfast. 
 **Maya**: Vitthal, serve breakfast to everyone. 
 Indravardhan: I will never drink milk. Maya, I have had diarrhea six times since morning. | Disgust |

Table 9: Examples of dialogues we queried ChatGPT along with their English translations and emotion labels recognised by ChatGPT. The target utterances are highlied with red colour.