# OpenReview forum: "From Multilingual Complexity to Emotional Clarity: Leveraging Commonsense to Unveil Emotions in Code-Mixed Dialogues"
_EMNLP/2023/Conference — EMNLP 2023 Main_

### Official Review · Reviewer_FCsy · 2023-08-05

**Soundness:** 4

**Excitement:**

4: Strong: This paper deepens the understanding of some phenomenon or lowers the barriers to an existing research direction.

**Missing References:**

- l.135-7: Could you provide some more details on these DL methods for ERC? Which general trends are there?

**Paper Topic And Main Contributions:**

The authors of this paper present the first dataset for emotion recognition in code-mixed conversations, called E-MaSaC. They also propose COFFEE, a method to extract commonsense knowledge from English knowledge graphs (not trivial given the code-mixed data) and fuse this knowledge with dialogue context.

**Questions For The Authors:**

- l.135-7: Could you provide some more details on these DL methods for ERC? Which general trends are there?
- l.199-200:
(a) What is the average length of these conversations and utterances? Could you also report standard deviation for average conversation length?
(b) In the introduction (l. 113), you state that you work on multi-party conversations, while here you state that you work on dialogues. Which of the two is true?
- Section 4:
(a) Could you add a table in which you calculate annotator agreement for each emotion label seperately?
(b) How did you handle instances where multiple emotion labels were applicable for a single utterance? Which label did annotators have to choose in such cases?
(c) Footnote 4: How were the annotators remunerated? Per hour (focus on quality) or per instance (focus on quantity)?
(d) l.221-2: how rare? Can you give an indication of how often such discrepancies occurred relative to the total number of instances to label?
(e) Why was Kriprendorff’s Alpha calculated on annotator pairs instead of all annotators together if all annotators labelled the same instances?
(f) Were IAA scores calculated before or after the annotators solved their discrepancies? If after, what was the IAA before?
- l.253-60: DUB: Did you add [SEP] tokens between utterances to signal where one utterance ends and another one starts? If not, how were utterances joint?
- Section 6.5.2 on subjective evaluation: Could you, similarly, look at an example in which the RoBERTa model with COFFEE approach performs worse than the RoBERTa model without COFFEE approach. What might be the cause?

**Reasons To Accept:**

- The authors have created the first dataset for emotion recognition in conversation (ERC) in a code-mixed setup. So far, ERC has not been investigated in a code-mixed setting and this novel resource will help future research on code-mixing.
- The authors also present COFFEE, a method to extract commonsense knowledge from English knowledge graphs and use it in a code-mixing setup. The validity of this model is shown through various quantitative and qualitative analyses.
- The paper is well-written and presents not only an interesting, but highly relevant contribution to the field of ERC and code-mixing.

**Reasons To Reject:**

- Some details on the dataset are missing (see questions for authors).
- Although the model COFFEE improves upon the baselines, the improvements are relatively small.
- The section on ChatGPT (Section 6.6) is somewhat out of line with the rest of the paper and seems to have been added to an already nicely structured project. Although such an addition is valuable, the current addition does not fit the structure of the paper. If you incorporate this model, it should be included earlier throughout the experimental section, not at the end after all analyses. Moreover, performance is not clearly reported (metric, number of instances involved).

**Reproducibility:**

4: Could mostly reproduce the results, but there may be some variation because of sample variance or minor variations in their interpretation of the protocol or method.

**Reviewer Confidence:**

4: Quite sure. I tried to check the important points carefully. It's unlikely, though conceivable, that I missed something that should affect my ratings.

**Typos Grammar Style And Presentation Improvements:**

- Abstract (l.13-5): you seem to claim that your innovation is integrating commonsense knowledge with dialogue context, while this has already been studied before. Obviously, there is novelty to this approach due to the code-mixing setup, but could you rephrase this part slightly to avoid confusion?
- l.183-4: Ekman's model consists of 6 instead of 7 basic emotions.
- Figure 3: There are still capitals in the text of the preprocessing step
- l. 307: Refer -> We refer to
- l. 390-3: RoBERTa is also trained without next-sentence prediction loss, in contrast to BERT
- l.425: contributes -> contribute
- l.449-50: Could you summarize this detailed analysis in a sentence or two?
- Table 5: Could the order of attributes be changed to the order in Figure 4? (easier for comparison)
- l.508-24: You refer to the section on the ablation study the quantitative section, while the section prior to this (Section 6.3) is called "Quantitative analysis". Moreover, the confusion matrix in section 6.5.1 is also a quantitative analysis.

---

> ### Author Rebuttal · Authors · 2023-08-28
>
> Thank you for your valuable comments, which we address below.
>
> **(Performance gain by COFFEE)** We appreciate the concern regarding the performance gain. However, we would like to clarify a possible misunderstanding in this regard. In Table 3, we reported the experimental results as raw F1 scores without converting them to percentages. Consequently, an increase of 0.03 F1 score for the RoBERTa model signifies a 3% increase (from 41% to 44%). We have recently performed a T-test on our results to check the statistical significance of our performance gain and obtained a p-value of 0.0321 which, being less than 0.05, makes our results statistically significant.
>
> **(ChatGPT section)** The section on ChatGPT was incorporated to highlight future research directions. We wanted to focus on the fact that while ChatGPT solves many problems and responds appropriately to variable queries, the research in the NLP domain is still far from complete. Consequently, we highlighted one drawback of the LLM related to how it handles a mix of languages in text since it was in line with our work. However, due to the associated cost with ChatGPT or the GPT3 API, we couldn't get the predictions for our complete test set. Thus, we couldn't provide comparable metrics for the complete set. Consequently, we randomly selected a few instances from the test set and queried them into ChatGPT. Specifically, we queried four instances and showed their results in the Appendix (Figure 6 and Table 9).
>
> **(More data statistics)** We summarize the requested dataset statistics here. We will add these statistics in the manuscript, too.
> 1.  ***Average length of utterances:*** 31.55 words (Mentioned in Table 1 of the main paper)
> 2.  ***Average length of conversations:*** 24.76 utterances
> 3.  ***Standard deviation of conversation length:*** 14.55
>
> Please note that we work on multi-party conversation and use the terms 'dialogue' with 'conversation' interchangeably. Specifically, the term 'dialogue' refers to a conversation between two or more people in our study. We will mention this explicitly in the manuscript and make it consistent.
>
> **(More details on annotations)** Please find the clarifications regarding emotion annotations below. We will add these to the manuscript.
> 1. ***Annotator agreement for individual emotion label:***
>
> | Anger | Contempt | Disgust | Fear | Joy | Neutral | Sadness | Surprise |
> |-------|----------|---------|------|-----|---------|---------|----------|
> | 0.87  |  0.81    |  0.81   | 0.82 |0.93 |  0.89   |  0.87   |   0.83   |
>
> 2. ***Multiple emotion labels for a single utterance:*** While annotating the instances, the annotators were asked to focus on the most prominent emotion in the utterance and assign that to the entire statement. We did not encounter any instance in our data where more than one emotion played a *crucial* part. Thus, our prominent emotion strategy worked for our data.
>
> 3. ***Annotator remuneration:*** The annotators were linguistics experts. They were paid as per the institute rule.
>
> 4. ***Annotation discrepancies:*** Out of the 11440 total utterances, there were only 43 utterances where the annotators had a disagreement over emotion labels. Most of them were one word utterances, and the discrepancies were resolved after a discussion among the annotators.
>
> 5. ***IAA calculation:*** The IAA score was calculated after resolving the discrepancies. Since the number of discrepancies is low with respect to the total number of utterances, the score before discrepancy resolution is similar to the final score of 0.85.
>
> **(Utterance separator)** Yes, we used the [SEP] token to join different utterances.
>
> **(Subjective evaluation)** While there are several instances for which COFFEE outperforms the non-CS approach, there are a few samples where the opposite is observed. We show one such example in the table below.
>
> |   |   Speaker   | Utterance | Gold Emotion | Emotion w/o CS | Emotion w CS |
> |---|-------------|-----------|---------|------|--------|
> |u1 |    Maya     | Vishal ji I am so sorry, Monisha ki jidh thi ki hum log ish tara se corridor me cricket khele toh. *(Vishal ji m so sorry, it was Monisha's insistence that we should play cricket in the corridor like this.)* | neutral | neutral | sad |
> |u2 | Vishal Guha | Nehi nehi koi baat nehi. Hamari cricket career ki suruwat bhi corridor se hi huyi hai. *(No no no problem. Our cricket career also started from the corridor itself.)* | neutral | joy | joy |
> |u3 |    Indu     | Accha. Chaliye chaliye. Chaliye vishal ji game suru kijiye chaliye. *(Good. Let's go let's start the game vishal ji.)* | joy | joy | joy |
> |u4 | Vishal Guha | Bat kaun karega? *(Who will bat?)* | neutral | neutral | neutral |
> |u5 |   Monisha   | Mai. *(Me.)* | joy | joy | surprise |
> |u6 | Vishal Guha | Chalo. *(Let's go.)* | neutral | joy | neutral |
>
> The instances where COFFE fails to capture the correct emotions might be due to the fact that the commonsense acts as noise instead of a helper and adds confusion to the system.
>
> **(Grammar, style, and presentation improvements)** Thank you for pointing these out. We will add the suggested edits to the manuscript.

---

### Official Review · Reviewer_m45s · 2023-08-06

**Soundness:** 4

**Excitement:**

4: Strong: This paper deepens the understanding of some phenomenon or lowers the barriers to an existing research direction.

**Paper Topic And Main Contributions:**

The author created a dataset of English-Hindi multiparty conversations with emotional annotations for each turn. They introduced a new method to extract commonsense knowledge from Hindi words from a pretrained English commonsense knowledge graph.

**Questions For The Authors:**

A. What is the performance when only considering the English words in the conversation turn for commonsense?

**Reasons To Accept:**

Their dataset would encourage the study of emotions in code-mixed conversations.

**Reasons To Reject:**

* Translating only parts of each conversation turn may lead to a loss of meaning. As a result, the commonsense knowledge extracted might be inaccurate.

**Reproducibility:**

4: Could mostly reproduce the results, but there may be some variation because of sample variance or minor variations in their interpretation of the protocol or method.

**Reviewer Confidence:**

2: Willing to defend my evaluation, but it is fairly likely that I missed some details, didn't understand some central points, or can't be sure about the novelty of the work.

---

> ### Author Rebuttal · Authors · 2023-08-28
>
> Thank you for your valuable comments, which we address below.
>
> **(Comparison with commonsense ERC models)** Since the existing commonsense ERC architectures were curated only for English, we did not consider them in our comparative systems. However, considering the criticality of the reviewer's suggestion, we have run the suggested baselines on our data and provided the results in the table below.
>
> |   Model   | Anger | Contempt | Disgust | Fear | Joy | Neutral | Sadness | Surprise | Wt. F1 |
> |-----------|-------|----------|---------|------|-----|---------|---------|----------|--------|
> | KET [1]   | 0.14  |   0.11   |  0.09   | 0.0  |0.34 |  0.41   |  0.08   |  0.19    |  0.28  |
> | COSMIC [2]| 0.21  |   0.18   |  0.15   | 0.03 |0.39 |  0.49   |  0.13   |  0.27    |  0.34  |
> | **Ours**      | **0.29**  |   **0.24**   | **0.18**   | **0.10** |**0.49** | **0.61**   |  **0.18**   |  **0.34**    |  **0.44**  |
>
> We can observe from the table that the existing strategies of commonsense fusion perform poorly when compared with the COFFEE method. The loss in performance can be attributed to two aspects of the comparative system --
> 1. KET [1] uses NRC_VAD [3], which is an English-based lexicon containing VAD scores, i.e., valence, arousal, and dominance scores, to gather words for which knowledge is to be retrieved. Since our input is code-mixed with the matrix language as Hindi, using only the English terms makes the KET approach ineffective.
> 2. Although COSMIC [2] uses the COMET [4] graph, it uses the raw representations obtained from the commonsense graph and concatenates them with the utterance representations obtained from the GRU architecture. Since we use the generated natural language commonsense with the smart fusion method, we hypothesize that our model is able to capture and utilize this knowledge effectively.
>
> *[1] Zhong, Peixiang, Di Wang, and Chunyan Miao. Knowledge-enriched transformer for emotion detection in textual conversations. EMNLP-IJCNLP, 2019.*
>
> *[2] Deepanway Ghosal, Navonil Majumder, Alexander Gelbukh, Rada Mihalcea and Soujanya Poria. COSMIC: COmmonSense knowledge for eMotion Identification in Conversations. EMNLP (Findings), 2020.*
>
> *[3] Saif Mohammad. Obtaining Reliable Human Ratings of Valence, Arousal, and Dominance for 20,000 English Words. ACL, 2018.*
>
> *[4] Antoine Bosselut, Hannah Rashkin, Maarten Sap, Chai tanya Malaviya, Asli Celikyilmaz, and Yejin Choi. COMET: Commonsense transformers for auto640 matic knowledge graph construction. ACL, 2019.*
>
> **(Performance gain by COFFEE is minimal)** We appreciate the concern regarding the performance gain. However, we would like to clarify a possible misunderstanding in this regard. In Table 3, we showed the experimental results as the raw F1 scores without converting them to percentages. Consequently, an increase of 0.03 in the F1 score for the RoBERTa model signifies a 3% increase (from 41% to 44%). We have recently performed a T-test on our results to check the statistical significance of our performance gain and obtained a p-value of 0.0321 which, being less than 0.05, makes our results statistically significant.
>
> **(Loss of knowledge due to translation)** We used the googletrans python library to translate sentences into English. This library provides the free use of Google translate API. To ensure the integrity of our dataset, we diligently performed manual checks. However, the COMET commonsense network can take input with a maximum sequence length of 17 tokens, which is less than most of the sequence lengths in our input since our average sequence length, as mentioned in Table 1 of the paper, is 31.55 tokens. To address this, we removed the stop word to adjust the sequence length while retaining information. This strategy's simplicity and effectiveness guided our choice.
>
> To reinforce the quality of translations, we show some examples in the table below. We can observe that the translated text retains its quality, preserving the relevance of the extracted commonsense knowledge to the original input.
>
> | Input (After preprocessing) | Translation | Commonsense knowledge |
> |-----------------------------|-------------|-----------------------|
> | monisha तुम्हारे फटे हुए कुर्ते rosesh थोड़ा फरक | monisha your torn kurta rosesh slightly different | xWant: to be unique; oReact: wear a new one |
> | indravadan, कितने सालो joke मार रहे. कोई हँसा आज तक? | Indravadan, how many years have you been joking? Anyone laughed today? | xWant: to be funny; oReact: laughs |
> | एक minute एक minute kismi. इसका मतलब railway station जाकर सोने सोच | One minute one minute kismi. This means thinking of going to the railway station and sleeping. | xWant: one minute; oReact: get to sleep |
>
> Admittedly, though the translation achieves a decent level of accuracy, we acknowledge the presence of nuances that warrant further enhancement. However, the intricate realm of language translation is a domain distinct from the focus of our current paper, and thus, we do not discuss it further in the work.
>
> **(English-only performance)** In code-mixing, input amalgamates two or more languages, often with one language being the dominant one, called the matrix language, while others act as embedding languages. The foundation of grammatical structure comes from the matrix language (Hindi in our case), and solely relying on the embedding language (English in our case) can lead to a decline in the model's performance. On the flip side, the embedding language plays a vital role in capturing accurate contextual details within the input. Therefore, confining ourselves to only the matrix language should also result in a drop in performance.
>
> The table below shows the performance of the COFFEE methodology, using the RoBERTa model, when we use only English (the embedding language) and only Hindi (the matrix language) in our input.
>
> |  Model  | Anger | Contempt | Disgust | Fear | Joy | Neutral | Sadness | Surprise | Wt. F1 |
> |---------|-------|----------|---------|------|-----|---------|---------|----------|--------|
> | English | 0.11  |   0.09   |  0.01   | 0.0  |0.16 |   0.24  |  0.02   |   0.11   |  0.16  |
> |  Hindi  | 0.2   |   0.15   |  0.12   | 0.02 |0.36 |   0.53  |  0.12   |   0.29   |  0.35  |
> |  **Both**   | **0.29**  |   **0.24**   |  **0.18**   | **0.10** |**0.49** |   **0.61**  |  **0.18**   |   **0.34**   |  **0.44**  |
>
> We observe the same pattern as discussed above, where the usage of only embedding language (English only) deteriorates the model performance extensively, while the sole use of matrix language (Hindi only) also hampers the performance when compared to the system that uses both the languages.

---

### Official Review · Reviewer_QymD · 2023-08-12

**Typos Grammar Style And Presentation Improvements:** Minor Grammatical punction errors.
**Soundness:** 4

**Excitement:**

4: Strong: This paper deepens the understanding of some phenomenon or lowers the barriers to an existing research direction.

**Missing References:**

One reference that closely aligns with the manuscript to address Knowlege Noise and Relevance of the KG (Quantitative  and Qualitative) :
 “K-LM: Knowledge Augmenting in Language Models Within the Scholarly Domain” (DOI: 10.1109/ACCESS.2022.3201542)

**Paper Topic And Main Contributions:**

The manuscript “From Multilingual Complexity to Emotional Clarity: Leveraging Commonsense to Unveil Emotions in Code-Mixed Dialogues” presents an approach that integrates commonsense information with dialogue context to interpret emotions better by means of a pipeline based on a knowledge graph to use relevant knowledge.

**Questions For The Authors:**

A. Which resource is used to identify the word written in English as a Hindi word to convert it to Devanagari?
B. How do you determine the relevance of Knowledge Graphs (in this case, COMET) with respect to the dataset?
C. Is the proposed model domain agnostic?
D. If not, what measures can be implemented to avoid Knowledge Noise?

**Reasons To Accept:**

The work is novel and new from the perspective that in India, Hindi and English are fused in a way, that it is common practice to use English words while speaking Hindi, and it does not follow any structure. Therefore, it makes communication unusual and complicated for machines to comprehend. This work is a step toward addressing the challenges in code-mixed language in multilingual settings. The world also contributes—a dataset that is helpful for the research community, and unfolds another use case for ChatGpt.


**Reasons To Reject:**

Since the work is novel in nature, I do not see any strong reason to reject the manuscript.

**Reproducibility:**

4: Could mostly reproduce the results, but there may be some variation because of sample variance or minor variations in their interpretation of the protocol or method.

**Reviewer Confidence:**

5: Positive that my evaluation is correct. I read the paper very carefully and I am very familiar with related work.

---

> ### Author Rebuttal · Authors · 2023-08-28
>
> Thank you for your valuable comments, which we address below.
>
> **A. (Resource to identify Hindi words)** We used the Textcat module of the NLTK library to identify languages. It uses the TextCat algorithm proposed by Cavnar et al. [1].
>
> *[1] Cavnar, W. B. and J. M. Trenkle, "N-Gram-Based Text Categorization".*
>
> **B. (Why COMET)** Since we had open domain chit-chat-based conversations in our dataset, we wanted to apply a generic commonsense architecture that returned various characteristics of the input. Since COMET returned multiple attributes related to speakers and listeners, it fitted perfectly with our requirements. Moreover, we did a thorough literature review and found COMET to be a popular choice for the English emotion recognition task and, thus, decided to go forward with it.
>
> **C&D. (Is the model domain agnostic?)** We tested our model in the open-domain chit-chat setting, where the speakers can talk about *anything*. Consequently, we can say that the model works for the open domain category, making it domain-independent.
>
> **(Missing references and typos)** Thank you for pointing these out. We will add the suggested edits to the updated manuscript.

---

### Meta-Review · Area_Chair_X2xY · 2023-09-21

**Recommendation:** 5

**Metareview:**

The paper studies the task of Emotion Recognition in Conversations (ERC) in the code-mixed setting. In particular, it studies the commonsense-aided ERC task.

Largely, the paper (a) Proposes a dataset towards this task (b) Does a study of translating code-mixed sentences to English to perform commonsense inference, and (c) Provides comprehensive experiments.

Reviewers appreciated the presentation of the paper and its technical soundness. The questions raised during the discussion were aptly addressed by the authors. Furthermore, reviewers liked the fact that ERC is being studied in code-mixed setting and appreciated the promise to release the prepared dataset.

---

### Decision · Program_Chairs · 2023-10-07

**Decision:**

Accept-Main

**Comment:**

The paper studies the task of Emotion Recognition in Conversations (ERC) in the code-mixed setting. In particular, it studies the commonsense-aided ERC task.

Largely, the paper (a) Proposes a dataset towards this task (b) Does a study of translating code-mixed sentences to English to perform commonsense inference, and (c) Provides comprehensive experiments.

Reviewers appreciated the presentation of the paper and its technical soundness. The questions raised during the discussion were aptly addressed by the authors. Furthermore, reviewers liked the fact that ERC is being studied in code-mixed setting and appreciated the promise to release the prepared dataset.